# Influence of Active and Healthy Ageing on Quality of Life Changes: Insights from the Comparison of Three European Countries

**DOI:** 10.3390/ijerph18084152

**Published:** 2021-04-14

**Authors:** Alba Ayala, Carmen Rodríguez-Blázquez, Amaia Calderón-Larrañaga, Giorgi Beridze, Laetitia Teixeira, Lia Araújo, Fermina Rojo-Pérez, Gloria Fernández-Mayoralas, Vicente Rodríguez-Rodríguez, Víctor Quirós-González, Vanessa Zorrilla-Muñoz, María Silveria Agulló-Tomás, Oscar Ribeiro, Maria João Forjaz

**Affiliations:** 1Department of Statistics, University Carlos III of Madrid, 28903 Getafe, Spain; alba.ayala@isciii.es; 2Health Service Research Network on Chronic Diseases (REDISSEC), 28029 Madrid, Spain; jforjaz@isciii.es; 3University Institute of Gender Studies, University Carlos III of Madrid, 28903 Getafe, Spain; vzorrill@ing.uc3m.es (V.Z.-M.); msat@polsoc.uc3m.es (M.S.A.-T.); 4National Epidemiology Centre, Carlos III Institute of Health, 28029 Madrid, Spain; 5Network Center for Biomedical Research in Neurodegenerative Diseases (CIBERNED), 28031 Madrid, Spain; 6Aging Research Center, Department of Neurobiology, Care Sciences and Society, Karolinska Institutet, 17165 Stockholm, Sweden; amaia.calderon.larranaga@ki.se (A.C.-L.); giorgi.beridze@ki.se (G.B.); 7Institute of Biomedical Sciences Abel Salazar, University of Porto, 4050-313 Porto, Portugal; laetitiateixeir@gmail.com; 8Center for Health Technology and Services Research (CINTESIS), 4200-450 Porto, Portugal; liajaraujo@esev.ipv.pt (L.A.); oribeiro@ua.pt (O.R.); 9School of Education, Polytechnic Institute of Viseu, 3504-501 Porto, Portugal; 10Institute of Economics, Geography and Demography, Spanish National Research Council (IEGD-CSIC), 28037 Madrid, Spain; fermina.rojo@csic.es (F.R.-P.); gloria.fernandezmayoralas@csic.es (G.F.-M.); vicente.rodriguez@csic.es (V.R.-R.); 11Dirección de Planificación, Hospital Universitario 12 de Octubre, 28041 Madrid, Spain; victor.quiros@salud.madrid.org; 12Department of Education and Psychology, University of Aveiro, 3810-193 Aveiro, Portugal

**Keywords:** quality of life, aging, participation, cross-national, longitudinal, SHARE project

## Abstract

This study aimed to analyze the determinants of quality of life (QoL) in older people in three European countries (Portugal, Spain and Sweden). A sample of 7589 participants in waves 4 (2011) and 6 (2015) of the Survey on Health, Aging, and Retirement in Europe (SHARE) project, aged 50 and over and living in Portugal, Spain and Sweden, was included. The CASP-12 scale was used to measure QoL. A principal component analysis was performed to group preselected variables related to active and healthy ageing into the dimensions of health, social participation, and lifelong learning. A linear regression model was built using the change in CASP-12 scores over the 4-year follow-up as the dependent variable, including the interactions between country and each independent variable in the model. After four years, the average QoL increased in Portugal (difference = 0.8, *p* < 0.001), decreased in Spain (−0.8, *p* < 0.001), and remained constant in Sweden (0.1, *p* = 0.408). A significant country-participation component interaction (*p* = 0.039) was found. In Spain, a higher participation (β = 0.031, *p* = 0.002) was related to a higher QoL improvement at 4 years, but not in Sweden or Portugal. Physical health and emotional components (β = 0.099, *p* < 0.001), functional ability (β = 0.044, *p* = 0.023), and cognitive and sensory ability (β = 0.021, *p* = 0.026) were associated with QoL changes over time in all countries. The country-specific associations between health, social participation and QoL should be taken into account when developing public health policies to promote QoL among European older people.

## 1. Introduction

Ageing is a biological process that also involves individual, social, political, economic, and cultural challenges. In 2002, the concept of active ageing, defined as “the process of optimizing opportunities for health, participation, and security in order to enhance quality of life as people age”, was published and later updated by the International Longevity Center (ILC-BR) to include a fourth pillar, i.e., lifelong learning [1]. The World Health Organization (WHO) also defined healthy ageing in 2015 as “the process of developing and maintaining the functional ability that enables wellbeing in old age” [2]. Both frameworks capture the comprehensive life-course approach of the ageing process within a multidimensional perspective that combines both personal and behavioral circumstances, as well as contextual and environmental aspects [3]. In both cases, the final goal is to promote the quality of life (QoL) of older people, defined as “the individuals’ perception of their position in life in the context of the culture and value systems in which they live and in relation to their goals, expectations, standards and concerns” [4].

When revising the literature, health and social participation emerge as two of the most important dimensions of active and healthy ageing that stand out in older people’s opinions [5,6]. Moreover, both dimensions are important determinants of QoL in older age [7,8]. Several studies have found a non-linear relationship between QoL and age [9] and, thus, it is key to carry out longitudinal studies that analyze QoL trajectories in old age. In addition, QoL differs between individuals and contexts. Differences in culture, living conditions, healthcare resources and policies have a measurable effect on how European citizens age [8,10]. In fact, reports have described differences in the distribution of ageing conditions such as multimorbidity or frailty across European countries, with marked north/south and east/west gradients [11,12]. Some of these variations are especially marked when comparing data from northern and southern countries for health indicators related to mobility, such as walking speed and grip strength [13], loneliness [14], as well as the for levels of quality of life and levels of socio-economic status, with consistently lower scores in southern countries like Spain, Italy, and Greece [15].

This study will focus on two southern (Portugal and Spain) and one northern European (Sweden) country. In all three countries, the proportion of people aged 65 or more was already around one fifth of the population in 2019 [16], and the relative increase in the very old group (i.e., people aged 85 years or more) is projected to be among the highest in Europe. The importance of comparing these European countries lies in their different demographic evolution and projections. Although at the present time they are aging countries, they have been through different demographic stages. Thus, in the 1950s, Sweden along with Denmark and the Netherlands, had the longest life expectancy in the continent, while it was not until 2009 that Portugal made great progress [17]. At the present time, the three countries are slightly above the average in relation to the proportion of the elderly population. But demographic projections suggest that in 2050 Spain and Portugal will surpass Sweden (36.8%, 34.8%, 24.6%, respectively) [18]. This demographic reality, strongly supported by the increasing ageing of the population structure, must be among the parameters to design and implement public policies that guarantee the well-being of the elderly [18].

The purpose of this article was to analyze how health, social participation and lifelong learning indicators predict changes in QoL in older people, and to ascertain if such predictors differ from one country to another. Using data from the Survey of Health, Ageing and Retirement in Europe (SHARE), this study hypothesizes that there will be significant differences in QoL between two waves of the survey and in the three analyzed countries (Portugal, Spain and Sweden). In addition, we will explore the determinants of change in QoL in the three countries.

Most studies on QoL and ageing have focused on samples from a single country, and a unique time point, preventing drawing conclusions about ageing trajectories and changes in time. This knowledge gap is widened by the omission of a theoretical framework that would guide the selection of ageing indicators. The main contributions of this study are three-fold: use of longitudinal, cross-national data, and based on the heathy and active ageing framework, which allows for a comprehensive view of ageing. Its results will guide the development of social and health policies and programs to maintain and increase the QoL of the older population.

## 2. Materials and Methods

### 2.1. Study Design and Sample

This longitudinal study was based on representative samples of people aged 50 or more from Portugal, Spain and Sweden, and used data from wave 4 (W4, 2011) and wave 6 (W6, 2015) of the Survey of Health, Ageing and Retirement in Europe (SHARE) [19,20,21]. These two waves were chosen since they provided data about the measures of interest for the three selected countries. This survey collects information on social, economic, and health aspects of European older individuals through computer-assisted personal interviewing (CAPI). The fieldwork procedures in the two waves were revised and approved by the Ethics Council of the Max Planck Society. The SHARE survey carries out a rigorous procedure that minimizes the methodological differences among countries and guarantees an ex-ante harmonized cross-national design [19,22].

The data used in this study refer to the participants in Portugal, Spain, and Sweden who participated both in the fourth and the sixth wave (*n* = 1963, 3664, and 1962, respectively). This sample size is large enough to find statistically significant differences with an α = 0.05 and a statistical power of 80%, in a repeated-measures design [19].

### 2.2. Measurements

QoL was evaluated with an adapted version of the CASP-12 scale that has shown good psychometric properties in different countries [23,24,25,26]. The CASP-12 scale measures QoL in older individuals and comprises 12 items divided into four dimensions: control, autonomy, self-realization, and pleasure. Each item is scored on a 4-point Likert-type scale (from 1 = often to 4 = never, although items 4 and 7 to 12 are reversed). The total score ranges from 12 to 48, where high values indicate a better QoL.

Five experts in the field of ageing research reviewed the SHARE databases to identify indicators of health, social participation and lifelong learning, the main predictors related to active and healthy ageing examined in the present study. Along with security, these dimensions correspond to the pillars of the active ageing framework [1]. In addition, health and participation are the core of the healthy ageing model [2]. Disagreements were resolved by consensus, and variables with more than 10% of missing cases were discarded. These indicators were then grouped, using principal component analysis, into three main components for the health dimension (good physical and emotional health; functional ability; and cognitive and sensory ability), a single component for the social participation dimension (number and frequency of participation in voluntary/charity work, sport/social, political/community, read books and magazines, word or number games, played cards or games such as chess, during the last year), and another one for the lifelong learning dimension (number of years of education, how often they attended courses in the last year, and self-rated reading and writing skills). All components ranged from 0 to 100, so that a higher score indicated better health, higher social participation, and higher levels of lifelong learning. Further information on the principal component analysis and a description of the indicators included is provided in Appendix A.

The models also included socio-demographic covariates such as sex, age, marital status (with partner: married or living together; without partner: single, separated or divorced, and widowed), and current job situation (retired, employed, homemaker, and non-employed). Wealth, used as a socio-economic indicator, was calculated by summing the following variables: value of main residence, value of any other real estate; value of bank accounts, bond, stock and mutual funds; and value of share part of a business and value of cars, and by subtracting mortgage on main residence and financial liabilities [20,27].

### 2.3. Statistical Analysis

All analyses were conducted using calibrated longitudinal weights to account for unit non-response and make the sample representative of the total population aged 50+ in each country [28]. The analysis began with a descriptive study of all variables used in each wave and country. Country differences were analyzed using the chi-square test for categorical variables and analysis of variance (ANOVA) for numerical variables. Next, CASP-12 means were calculated for each categorical variable and principal components (dichotomized by the median into low and high), analyzing the differences per wave using Student’s *t*-test for paired data.

In addition, a multiple imputation using chained equations (MICE) strategy was devised to impute the CASP-12 in W6 due to a high percentage of missing values (29.6%). In order to calculate the model estimations, the five imputed datasets were combined using Rubin’s rule [29]. Out of the 7589 participants, 5107 had complete information on CASP-12 in W6. There was 4867 participants with completed data in all our study variables. We imputed 2123 cases, of which 1883 are imputed cases of CASP-12 in W6, and 240 cases with missing data in W4, in the rest of variables.

A linear regression model with ordinary least squares (OLS) estimators was constructed to analyze the variables associated with the change in QoL. The dependent variable was the CASP-12 difference between waves (W6–W4). The CASP-12 and components of health, participation and lifelong learning, all from W4, were included as potential predictors of between-individual variability in QoL changes. Models were adjusted for age, gender, marital status, current job situation, wealth, as well as baseline QoL. Interactions between each variable included in the model and country were additionally analyzed.

Finally, as a sensitivity analysis, we repeated our main analysis on the sample with missing values, taking into account the missing values in CASP-12 at wave 6 (i.e., complete case analysis). Analyses were run using the statistical program Stata 15.1 (StataCorp LLC, College Station, TX, USA).

## 3. Results

Out of those 7589 participants included in our study, 2482 (32.9%) had missing information on CASP-12 at W6 (Table 1). Those with missing information on CASP-12 at W6 (non-respondents hereafter) had a higher mean age (70.7 vs. 66.6 years, *p* < 0.001) than respondents and represented a higher percentage in Spain (34.1%, *p* = 0.027). Non-respondents had a higher percentage of people who were single (38.2%, *p* < 0.001), and a lower percentage of women (31.2%, *p* = 0.002). Mean baseline CASP-12 of non-respondents was also lower (34.2, *p* < 0.001), and they had lower scores in the health, participation, and learning components (all *p* < 0.001). The Appendix A shows the descriptive statistics of respondents and non-respondents in W6 stratified by country.

In W4, significant differences between countries were observed for almost all variables (Table 2). The mean QoL was highest in Sweden (38.5, standard deviation, SD = 5.4) and lowest in Portugal (31.9, SD = 5.1). Sweden also had significantly higher scores in all components.

As shown in Figure 1, the average QoL increased in Portugal after 4 years (W4 = 32.3, W6 = 33.1, *p* < 0.001), while the average QoL scores decreased in Spain (W4 = 36.0, W6 = 35.2, *p* < 0.001) and remained constant in Sweden (W4 = 39.2, W6 = 39.1, *p* = 0.408).

Table 3 shows the W4-W6 differences in QoL for each categorical variable. A significant decrease in QoL was observed in men (∆ = −0.3, *p* = 0.030), people living with a partner (∆ = −0.3, *p* = 0.008), retired people (∆ = −0.4, *p* = 0.001), and homemakers (∆ = −0.6, *p* = 0.011), individuals who reported higher levels of physical and emotional health (∆ = −0.5, *p* < 0.001), greater functional ability (∆ = −0.5, *p* < 0.001), higher cognitive and sensorial ability (∆ = −0.6, *p* < 0.001), lower participation (∆ = −0.3, *p* = 0.037) and greater scores of learning (∆ = −0.3, *p* = 0.007). The QoL also decreased in those with lower (∆ = −0.3, *p* = 0.046) and higher (∆ = −0.4, *p* = 0.027) wealth.

The final multiple linear regression model is shown in Table 4. The QoL change-related variables were physical health and emotional components (β = 0.099, *p* < 0.001), functional ability (β = 0.044, *p* = 0.023) and cognitive and sensory ability (β = 0.021, *p* = 0.026). A significant country-participation component interaction (*p* = 0.039) was identified in this model. An association was observed between a higher QoL at 4 years and a higher participation in Spain (β = 0.031, *p* = 0.002), which was not seen in Sweden (β = 0.000, *p* = 0.999) or Portugal (β = 0.003, *p* = 0.876, Figure 2). No other significant interactions by country were found. Sensitivity analysis with the non-imputed original CASP-12 variable for W6 showed similar results to those in the main analyses.

## 4. Discussion

This study showed that within-individual QoL changes are associated to different health components (physical, mental, and cognitive) as well as social participation activities in three European countries, which calls for a multidimensional understanding of wellbeing in old age.

In terms of cross-country comparisons, we found higher average QoL scores in Sweden and lower ones in Portugal. Studies about QoL in older Europeans alert about the disadvantage of Southern countries, which present lower QoL coefficients; within these, Portugal appears as the country with lowest levels of QoL [30]. According to the Active Ageing Index report [31], Spain and Portugal have a similar and much higher Gini index of inequality than Sweden, and the active ageing index is in fact higher in Sweden, whereas in Spain and Portugal the index is similar [31]. Additionally, the association between QoL and social factors, such as loneliness and social isolation, also differs between Sweden and Spain [32].

Our results showed that, between 2011 and 2015, QoL increased in Portugal and decreased in Spain. A comparative European study by country evaluated the changes in QoL between 2007 and 2012, and revealed the negative impact of the economic crisis on the standard of living, trust in institutions and other aspects of health conditions, although it did not find statistically significant changes in relation to the global QoL indicator, highlighting that “the most marked changes have been identified in the single groups of QoL rather than on the overall index QoL: this is because the opposite direction of some changes have tended to cancel out when we consider the overall average effect” [33].

The potential increase in QoL observed in Portugal could be associated with the social and economic situation of that period. Like other European countries, Portugal was affected by two major adverse events: the global financial crisis of 2008–2009 and the euro crisis of 2010–2012 [34]. These crises, marked by falling GDP, high unemployment rate and rising government debt, fostered an unequal distribution of power, status and resources impacting people’s lives, which in turn seriously threatened the older population’s mental wellbeing [35]. In Portugal, older people were of particular concern, since their low incomes were affected by austerity measures, including freezing of pensions and cuts in social benefits, increases in costs of health care, public transport and other daily expenses [36]. Recent studies on Portuguese older adults reported the importance of socio-economic position and social support from family, friends, or significant others in mitigating inequalities in QoL in later life [37], as well as the role of anxiety and depression [38]. However, a timid recovery of the economic situation has been reported for the 2014–2021 period [34].

In Spain, a worsening of the QoL has also been observed during the crisis between 2007 and 2011 [39]. Spain was one of the European countries where the consequences of the global economic crisis were most devastating. In fact, the recovery has not yet been completed and it is still causing worries to the population [40]. In this context, economically vulnerable older individuals were particularly affected by the rise in owned and rented housing prices and other conditions related to the properties [41,42]. In the Active Ageing Index report, Spain also scored worse than Portugal in employment in people aged 50 years or older, despite having scored higher than Portugal in participation, independent, healthy and safe living, and enabling capacity and environment components [31].

In our study, changes in QoL were independently associated with age, physical and emotional health, and functional ability components. Depression seems to be the main determinant of health-related QoL in older adults, while functional ability also limits QoL [43]. A study of community-dwelling older adults from Spain showed how people aged 78 years or older reported significantly lower QoL than their younger counterparts, and that some chronic health conditions, such as osteo-articular or mental health disorders, and disability have a large impact on health-related QoL and overall QoL [44]. Another study on Spanish community-dwelling older adults found depression to be the most important determinant for overall QoL, and the only significant one among all studied dimensions [45]. Depression is also a longitudinal predictor of QoL [38]. Higher life satisfaction, a global QoL indicator, is also associated with factors such as perception of the household’s economic position and satisfaction with the way people live, together with objective good health, subjective perception of health, absence of depression and high residential satisfaction [46,47].

Our study did not find a direct effect of wealth on QoL change, even though Portugal and Spain were influenced by the economic crisis during the study period. The economic crisis directly affected wealth, as well indirectly health and psychosocial variables. Financial means are one of the main active ageing determinants, because they facilitate independent and autonomous decision-making and involvement in activities [48]. These include the person’s and the household’s monthly income, welfare benefits, availability of owned housing or other assets, and/or family transfers [49,50]. Financial security is in fact the third dimension most often reported by older adults when defining their QoL, after health and family, although its relative importance diminishes when controlled by psychosocial and health variables [43,49], similarly to what was seen in our study.

This study showed an interaction between QoL, participation and country, as evidenced by the association of participation with increased QoL only in Spain. Policies aimed at increasing the QoL of older adults in Spain should promote social participation activities and enable places where these can occur. Previous studies have highlighted the relevant differences in QoL between countries participating in SHARE, with a clear north-south gradient. Socio-economic and welfare state indicators account for part of these differences, and some authors argue that social welfare models may mitigate or reverse the effects of poor physical health and disability on QoL in older adults [51,52]. Engaging in leisure activities and social participation is also associated with better QoL, and cross-cultural differences have also been previously found in the level and types of activities older adults perform [53,54]. To what extent disparities in socio-economic, health, and psychosocial indicators among Portugal, Spain and Sweden are contributing to the observed interaction is unknown, and thus further studies are needed to solve this question.

Certain limitations should be taken into account. Our study showed a loss of 32.9% participants from baseline to follow-up. Such a high percentage could lead to biased estimates. Another limitation is the relatively short duration of the QoL monitoring period in an observational study, and further, longer-lasting longitudinal studies would be needed to look for causal relationships. It would be very interesting to widen the geographical scope of the study, especially to include Eastern-Europe countries. Despite the limitations, our study also has other strengths. These include the use of a QoL scale validated in the three considered countries and a large dataset specifically designed to make comparisons between European countries. Also, the use of longitudinal data allowed the use of change in QoL as the main outcome.

## 5. Conclusions

In conclusion, determinants of QoL changes are multidimensional, thus calling for a multidisciplinary and multilevel approach. When comparing the QoL level of Sweden, Portugal and Spain, we observe a North-South gradient. Also, the observed change of QoL level in the study period was different by country and is probably related to a differential response to the economic crisis in the previous period. Our study showed that better physical, emotional and functional health in old age is related to an increase in QoL in Portugal, Spain, and Sweden, while a higher social participation is related to an increase in QoL only in Spain. Particularly in Spain, interventions that offer social participation activities and enable places for it, might be of great benefit for older adults’ QoL. This study suggests the importance of taking into account the influence that health and social participation have according to the context of each country when developing public health policies to promote QoL among European older people.

## Figures and Tables

**Figure 1 ijerph-18-04152-f001:**
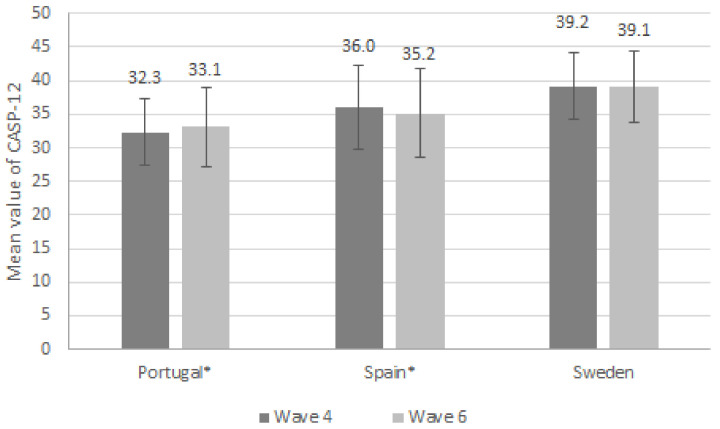
Means and standard QoL deviations (CASP-12) by country and wave. T-test for paired data (*n* = 5107). * significant differences at 5% significance level.

**Figure 2 ijerph-18-04152-f002:**
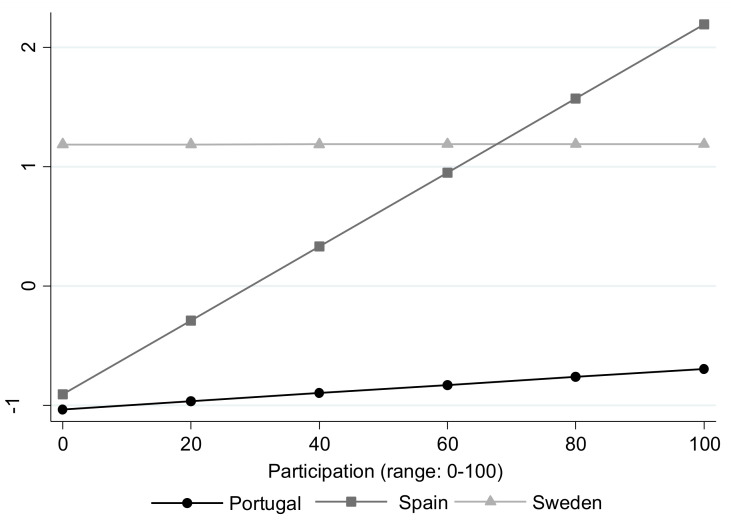
Country and participation interaction on the change in QoL. Linear regression model (CASP-12) adjusted by all predictor variables.

**Table 1 ijerph-18-04152-t001:** Baseline descriptive statistics of participants with complete and missing CASP-12 at W6 (unweighted).

	Participants with Complete CASP-12 at W6	Participants with Missing CASP-12 at W6	*p*-Value *
	*n* (Row %)	*n* (Row %)	
**Total**	5107 (67.3)	2482 (32.7)	
**Age**, mean (SD)	66.6 (9.3)	70.7 (11.6)	<0.001
**Country**			0.027
Portugal	1333 (67.9)	629 (32.1)	
Spain	2414 (65.9)	1250 (34.1)	
Sweden	1360 (69.3)	603 (30.7)	
**Sex**			0.002
Male	2238 (65.4)	1183 (34.6)	
Female	2869 (68.8)	1299 (31.2)	
**Marital status**			<0.001
Without partner	1055 (61.8)	652 (38.2)	
With partner	3922 (69.1)	1753 (30.9)	
**Current job situation**			<0.001
Retired	2547 (64.8)	1381 (35.2)	
Employed	1233 (74.3)	427 (25.7)	
Homemaker	865 (70.0)	371 (30.0)	
Other	436 (64.0)	245 (36.0)	
**CASP-12**, mean (SD)	35.9 (6.1)	34.2 (6.9)	<0.001
**Physical and emotional health**, mean (SD)	76.3 (11.6)	74.0 (13.6)	<0.001
**Functional ability**, mean (SD)	77.3 (10.0)	72.7 (15.7)	<0.001
**Cognitive and sensorial ability**, mean (SD)	50.2 (16.8)	47.8 (17.9)	<0.001
**Participation**, mean (SD)	28.8 (20.8)	25.7 (19.7)	<0.001
**Learning**, mean (SD)	45.4 (22.7)	41.9 (22.3)	<0.001
**Wealth (€)**, mean (SD)	272.2 (465.7)	251.8 (656.9)	0.255

* Chi-square and Student’s *t* tests.

**Table 2 ijerph-18-04152-t002:** Descriptive statistics of the study variables and differences by country in W4, complete cases (unweighted n, weighted percentage).

	Total Sample	Portugal	Spain	Sweden	*p*-Value *
	(*n* = 7589)	(*n* = 1962)	(*n* = 3664)	(*n* = 1963)	
	*n* (%)	*n* (%)	*n* (%)	*n* (%)	
**Age, mean (SD)**	67.0 (10.8)	66.5 (10.5)	67.0 (11.1)	67.6 (10.1)	0.042
**Gender**					0.539
Male	3419 (45.9)	862 (44.5)	1659 (45.9)	898 (47.5)	
Female	4130 (54.1)	1082 (55.5)	1985 (54.1)	1063 (52.5)	
**Marital status**					0.007
Without partner	1706 (30.7)	443 (26.1)	797 (31.0)	466 (34.6)	
With partner	5636 (69.3)	1498 (73.9)	2700 (69.0)	1438 (65.4)	
**Current job situation**					≤0.001
Retired	3928 (43.5)	1091 (54.3)	1496 (38.0)	1341 (56.2)	
Employed	1641 (25.9)	432 (21.9)	683 (23.9)	526 (39.7)	
Homemaker	1221 (19.3)	212 (12.6)	1000 (25.0)	9 (0.4)	
Other	675 (11.4)	182 (11.2)	433 (13.1)	60 (3.7)	
**CASP-12**, mean (SD)	35.5 (6.5)	31.9 (5.1)	35.7 (6.6)	38.5 (5.4)	≤0.001
**Physical and emotional health**, mean (SD)	76.0 (12.4)	74.5 (12.8)	75.7 (12.7)	79.0 (9.2)	≤0.001
**Functional ability**, mean (SD)	75.8 (12.0)	74.4 (12.0)	75.3 (11.3)	79.7 (13.8)	≤0.001
**Cognitive and sensorial ability**, mean (SD)	49.3 (17.1)	44.4 (17.3)	48.0 (16.3)	60.8 (15.8)	≤0.001
**Participation**, mean (SD)	25.7 (19.8)	19.6 (17.3)	23.1 (18.4)	46.5 (16.0)	≤0.001
**Lifelong Learning**, mean (SD)	42.6 (20.9)	33.5 (18.2)	39.1 (17.2)	70.2 (16.7)	≤0.001
**Wealth (€000 s)**, mean (SD)	273.3 (609.2)	143.4 (184.8)	294.5 (697.4)	338.9 (494.7)	≤0.001

* Chi-square and ANOVA tests.

**Table 3 ijerph-18-04152-t003:** Means and standard QoL deviations (CASP-12) by explanatory variables for each wave (imputed cases).

	Wave 4	Wave 6	*p*-Value *
	Mean (SD)	Mean (SD)	
**Sex**			
Male	36.7 (5.7)	36.4 (6.1)	0.030
Female	35.3 (6.3)	35.1 (6.7)	0.094
**Marital status**			
Without partner	35.1 (6.3)	35.0 (6.7)	0.655
With partner	36.1 (6.0)	35.8 (6.4)	0.008
**Current job situation**			
Retired	35.9 (6.1)	35.5 (6.4)	0.001
Employed	38.0 (5.3)	38.2 (5.6)	0.364
Homemaker	34.0 (6.3)	33.4 (6.4)	0.011
Other	33.8 (6.0)	34.0 (6.6)	0.481
**Physical and emotional health**			
Low (≤78)	33.0 (5.8)	33.0 (6.5)	0.524
High (>78)	38.5 (5.1)	38.0 (5.5)	<0.001
**Functional ability**			
Low (≤78)	33.1 (6.1)	33.3 (6.7)	0.176
High (>78)	38.0 (5.1)	37.5 (5.7)	<0.001
**Cognitive and sensorial ability**			
Low (≤48)	33.2 (5.9)	33.4 (6.5)	0.190
High (>48)	38.4 (5.1)	37.8 (5.7)	<0.001
**Participation**			
Low (≤22)	33.9 (6.0)	33.6 (6.5)	0.037
High (>22)	37.7 (5.6)	37.5 (5.9)	0.082
**Lifelong learning**			
Low (≤40)	33.5 (5.9)	33.3 (6.4)	0.383
High (>40)	38.0 (5.4)	37.7 (5.7)	0.007
**Wealth**			
Low (≤150€)	34.4 (6.1)	34.1 (6.4)	0.046
High (>150€)	37.5 (5.7)	37.1 (6.1)	0.027

* Student paired *t*-test. Levels for the physical and emotional health, functional ability, cognitive and sensorial ability, participation and learning components and wealth variable are based on the median.

**Table 4 ijerph-18-04152-t004:** Multiple linear regression of changes in QoL (CASP-12) between W4 and W6.

	Imputed CASP-12 for W6		Original CASP-12 for W6 (Complete Case Analysis)	
Variable (Reference)	β (CI 95%)	*p*-Value	β (CI 95%)	*p*-Value
**CASP-12 W4**	−0.682 (−0.735; −0.629)	≤0.001	−0.753 (−0.802; −0.704)	≤0.001
**Country** (Sweden)#participation				
Spain	0.039 (0.014; 0.064)	0.002	0.031 (0.006; 0.056)	0.025
Portugal	0.001 (−0.046; 0.048)	0.956	0.003 (−0.044; 0.050)	0.891
**Country** (Sweden)				
Spain	−2.420 (−3.669; −1.171)	≤0.001	−2.096 (−3.56; −0.632)	0.010
Portugal	−1.880 (−3.360; −0.400)	0.013	−2.220 (−4.164; −0.276)	0.040
**Age**	−0.051 (−0.088; −0.014)	0.008	−0.060 (−0.099; −0.021)	0.006
**Sex** (Male)				
Female	−0.439 (−1.031; 0.153)	0.146	−0.377 (−0.99; 0.236)	0.236
**Marital status** (Without partner)				
With partner	−0.379 (−1.063; 0.305)	0.278	−0.099 (−0.781; 0.583)	0.778
**Current job status** (Retired)				
Employed	−0.186 (−0.988; 0.616)	0.650	−0.255 (−0.992; 0.482)	0.499
Homemaker	−0.386 (−1.101; 0.329)	0.290	−0.432 (−1.143; 0.279)	0.239
Other	−0.593 (−1.642; 0.456)	0.268	−0.534 (−1.436; 0.368)	0.246
**Physical and emotional health component**	0.099 (0.066; 0.132)	≤0.001	0.096 (0.067; 0.125)	≤0.001
**Functional ability component**	0.044 (0.005; 0.083)	0.023	0.057 (0.026; 0.088)	0.001
**Cognitive and sensorial ability component**	0.021 (0.001; 0.041)	0.026	0.034 (0.014; 0.054)	0.003
**Participation component**	0.000 (−0.020; 0.020)	0.978	0.000 (−0.027; 0.027)	0.999
**Learning component**	0.015 (−0.005; 0.035)	0.146	0.019 (−0.005; 0.043)	0.157
**Wealth**	0.208 (−0.076; 0.492)	0.152	0.200 (−0.055; 0.455)	0.130
**Constant**	16.971 (12.408; 21.534)	≤0.001	18.413 (13.934; 22.892)	≤0.001
**Number of observations**	6990		4867	
**R^2^ (average over 5 imputations)**	0.333		0.311	

β: adjusted beta regression coefficient; CI 95%: confidence interval at 95% level; R^2^: determination coefficient.

## Data Availability

Data supporting our findings is available to the scientific community free of charge at www.share-project.org (accessed on 13 April 2021).

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
