# Peer review of "Influence of Active and Healthy Ageing on Quality of Life Changes: Insights from the Comparison of Three European Countries"

_ijerph, 2021, doi:10.3390/ijerph18084152_

Round 1

Reviewer 1 Report

The authors have done a good job. The structure, methods, data used are all qualified.

I only have one concern: please try to add knowledge gap, contributions and paper structure in the last paragraph of introduction section.

Please refer to this paper for your use: https://doi.org/10.3390/healthcare9010072

Author Response

The authors have done a good job. The structure, methods, data used are all qualified.

Answer: We thank the reviewer for this comment.

I only have one concern: please try to add knowledge gap, contributions and paper structure in the last paragraph of introduction section.

A last paragraph was added with the requested information. The paper structure is now presented in the before-last paragraph:

“Using data from the Survey of Health, Ageing and Retirement in Europe (SHARE), this study hypothesizes that there will be significant differences in QoL between two waves of the survey and in the three analyzed countries (Portugal, Spain and Sweden). In addition, we will explore the determinants of change in QoL in the three countries.

Most studies on QoL and ageing have focused on samples from a single country, and a unique time point, preventing drawing conclusions about ageing trajectories and changes in time. This knowledge gap is widened by the omission of a theoretical framework that would guide the selection of ageing indicators. The main contributions of this study are three-fold: use of longitudinal, cross-national data, and based on the heathy and active ageing framework, which allows for a comprehensive view of ageing. Its results will guide the development of social and health policies and programs to maintain and increase the QoL of the older population.”

Please refer to this paper for your use: https://doi.org/10.3390/healthcare9010072

Answer: According to the reviewer’s suggestion and the recommended paper, we have added the following text in the before-last paragraph of the Introduction:

“Using data from the Survey of Health, Ageing and Retirement in Europe (SHARE), this study hypothesizes that there will be significant differences in QoL between two waves of the survey and in the three analyzed countries (Portugal, Spain and Sweden). In addition, we will explore the determinants of change in QoL in the three countries.”

Reviewer 2 Report

Authors aimed to identify the strongest predictors of quality of life (QoL) in two cohorts from the Survey on Health, Aging, and Retirement in Europe (SHARE) project which included a sample of older adults from Portugal, Spain and Sweden. QoL was measured by responses to the CASP-12 scale. Authors performed a principal component analysis to group variables previously identified and agreed upon by a team of experts in the field into three dimensions including health, social participation and lifelong learning. Then, using the grouped components, authors built a linear regression model with change in CASP-12 scores over time as the dependent variable. Authors found a significant country x social participation component interaction. In Spain, higher participation was associated with greater improvement in QoL. The health components (physical and emotional health, functional ability, and cognitive and sensory ability) were all associated with QoL changes in all three countries.

Introduction (minor comments): Authors introduce, explain and justify most of their primary variables well. Authors introduce the importance of lifelong learning by citing the International Longevity Center’s updated publication, following this citation with a study (similar to those the authors’ cited that highlight walking speed and grip strength as indications of mobility which vary across northern and southern European countries) would further strengthen the justification for including lifelong learning in their analyses.

Materials and Methods (minor comments): It may be helpful to include a short sentence clarifying why Portugal, Spain and Sweden were chosen for this study. It seems as though the authors chose these countries because two are southern and one is northern (given the authors previously state a variability in QoL has been found between north-south countries) and wave 4 and wave 6 provided data about the primary measures. If this is accurate perhaps no clarification is needed.

If the procedures used by the SHARE survey that minimize methodological differences among countries has been published, it may be helpful to cite that publication after stating “the SHARE survey carries out a rigorous procedure that minimizes the methodological differences….” [line 88].

The additional “12” on line 135 seems erroneous

Results (minor comments): The sentence in the results starting on line 156 is long and reports a lot. It may be easier to read and digest if broken into two summary sentences rather than one. The results reported on line 158 are confusing, (“of the single people”), does this mean the following results are specific to the portion of the sample who are single or is this perhaps a typo?

Discussion (minor comments): The authors make important points about the economic and social events which impacted the QoL in Portugal I suggest explicitly stating these events likely impacted (decreased) the initial W4 QoL scores. It may help keep the reader focused on the point the authors are making.

The sentence beginning on line 235, “In Spain, QoL may have fallen because…” is redundant as it states, “…as it was one of the European countries where the consequences of the crisis of the global economic crisis…” (line 237-8).

Line 242, it is unclear how “environment” and “surroundings” denote two distinct ideas in this context.

Do the authors think the determinants of QoL may have played a role in how Spain and Portugal responded differently to the economic crisis? This point may be beyond the scope of this paper, however, if authors have any thoughts on how QoL may impact response and recovery to crises I believe including these thoughts in the discussion would strengthen the impact and implications of this manuscript.

Line 255 parentheses are used rather than brackets for the reference.

Authors may want to further discuss potential reasons why they did not find a significant effect of socio-economic status considering the impact of the economic crisis in Portugal and especially in Spain. The author’s point on psychosocial and health variables relative importance is well taken although it stands to reason health and psychosocial variables may have also been impacted by the economic crisis and therefore may not have had as strong of a controlling effect on the impact of financial security. This may be worth exploring further.

Country and social participation interaction in Spain, what are the potential implications of this on policy specific to Spain?

Overall, authors wrote a thorough and thoughtful manuscript on an interesting and relevant data analysis project. Authors carried out thorough and responsible data analyses and wrote a clear and concise manuscript.

Author Response

Authors aimed to identify the strongest predictors of quality of life (QoL) in two cohorts from the Survey on Health, Aging, and Retirement in Europe (SHARE) project which included a sample of older adults from Portugal, Spain and Sweden. QoL was measured by responses to the CASP-12 scale. Authors performed a principal component analysis to group variables previously identified and agreed upon by a team of experts in the field into three dimensions including health, social participation and lifelong learning. Then, using the grouped components, authors built a linear regression model with change in CASP-12 scores over time as the dependent variable. Authors found a significant country x social participation component interaction. In Spain, higher participation was associated with greater improvement in QoL. The health components (physical and emotional health, functional ability, and cognitive and sensory ability) were all associated with QoL changes in all three countries.

Introduction (minor comments): Authors introduce, explain and justify most of their primary variables well. Authors introduce the importance of lifelong learning by citing the International Longevity Center’s updated publication, following this citation with a study (similar to those the authors’ cited that highlight walking speed and grip strength as indications of mobility which vary across northern and southern European countries) would further strengthen the justification for including lifelong learning in their analyses.

Answer: As explained in the Introduction, lifelong learning is a pillar of Active Ageing in the definition published by International Longevity Center (Kalache, 2015). Moreover, promoting the lifelong learning and literacy in older adults is one of the aspects of the functional ability component that guarantee the healthy ageing in the WHO framework (WHO, 2020). According to SHARE survey, the North-South gradient in QoL is influenced by educative level among other determinants. These reasons justify the inclusion of lifelong learning in the analysis.

Reference:

WHO- World Health Organization. (2020). Decade of healthy ageing: baseline report. Geneva: World Health Organization. Retrieved from https://www.who.int/publications/m/item/decade-of-healthy-ageing-baseline-report

Materials and Methods (minor comments): It may be helpful to include a short sentence clarifying why Portugal, Spain and Sweden were chosen for this study. It seems as though the authors chose these countries because two are southern and one is northern (given the authors previously state a variability in QoL has been found between north-south countries) and wave 4 and wave 6 provided data about the primary measures. If this is accurate perhaps no clarification is needed.

Answer: There was an interest in comparing Northern and Southern countries such as Sweden, Portugal and Spain according to the evidence of a gradient found in previous studies using SHARE data. The importance of comparing these European countries lies in their different demographic evolution and projections.

Moreover, the research team is composed by researchers from the three countries that could provide insight and draw conclusions based on the socio-economic and cultural contexts of the participating countries. Indeed, waves 4 and 6 provide data for the three countries and the primary measures.

The 3rd paragraph of the Introduction has been modified accordingly:

“The importance of comparing these European countries lies in their different demographic evolution and projections. Although at the present time they are aging countries, they have been through different demographic stages. Thus, in the 1950s, Sweden along with Denmark and the Netherlands, had the longest life expectancy in the continent, while it was not until 2009 that Portugal made great progress [17]. At the present time, the three countries are slightly above the average in relation to the proportion of the elderly population. But demographic projections suggest that in 2050 Spain and Portugal will surpass Sweden (36.8%, 34.8%, 24.6%, respectively) [18]. This demographic reality, strongly supported by the increasing ageing of the population structure, must be among the parameters to design and implement public policies that guarantee the well-being of the elderly [18]”

If the procedures used by the SHARE survey that minimize methodological differences among countries has been published, it may be helpful to cite that publication after stating “the SHARE survey carries out a rigorous procedure that minimizes the methodological differences….” [line 88].

 Answer: The reference has been included:

“The SHARE survey carries out a rigorous procedure that minimizes the methodological differences among countries and guarantees an ex-ante harmonized cross-national design [19, 22].”

The additional “12” on line 135 seems erroneous

 Answer: It has been deleted, we apologize for the mistake.

Results (minor comments): The sentence in the results starting on line 156 is long and reports a lot. It may be easier to read and digest if broken into two summary sentences rather than one.

Answer: The sentence has been split so it is now easier to read and understand:

“Out of those 7589 participants included in our study, 2482 (32.9%) had missing information on CASP-12 at W6 (Table 1). Those with missing information on CASP-12 at W6 (non-respondents hereafter) had a higher mean age (70.7 vs. 66.6 years, p<0.001) than respondents and represented a higher percentage in Spain (34.3%, p=0.027). Non-respondents had a higher percentage of people who were single (38.2%, p<0.001), and a lower percentage of women (31.5%, p=0.002). Mean baseline CASP-12 of non-respondents was also lower (34.2, p<0.001), and they had lower scores in the health, participation, and learning components (all p<0.001).”

The results reported on line 158 are confusing, (“of the single people”), does this mean the following results are specific to the portion of the sample who are single or is this perhaps a typo?

Answer: The sentence has been re-written for higher clarity. Now it reads:

“Non-respondents had a higher percentage of people who were single (38.2%, p<0.001), and a lower percentage of women (31.5%, p=0.002).”

Discussion (minor comments): The authors make important points about the economic and social events which impacted the QoL in Portugal I suggest explicitly stating these events likely impacted (decreased) the initial W4 QoL scores. It may help keep the reader focused on the point the authors are making.

Answer:  we have added a selection of major points associated with the crisis, and now it reads:

“These crises, marked by falling GDP, high unemployment rate and rising government debt fostered an unequal distribution of power, status and resources impacting people’s lives, which in turn seriously threatened the older population’s mental wellbeing [35].”

The sentence beginning on line 235, “In Spain, QoL may have fallen because…” is redundant as it states, “…as it was one of the European countries where the consequences of the crisis of the global economic crisis…” (line 237-8).

Answer: This sentence has been corrected, and it now reads:

In Spain, a worsening of the QoL has also been observed during the crisis between 2007 and 2011 [39]. Spain was one of the European countries where the consequences of the global economic crisis were most devastating.

Line 242, it is unclear how “environment” and “surroundings” denote two distinct ideas in this context.

Answer: These terms were now deleted, as they might be confusing.

Do the authors think the determinants of QoL may have played a role in how Spain and Portugal responded differently to the economic crisis? This point may be beyond the scope of this paper, however, if authors have any thoughts on how QoL may impact response and recovery to crises I believe including these thoughts in the discussion would strengthen the impact and implications of this manuscript.

Answer: Thank you very much for your suggestions. This is an interesting, complex issue, for which a different study would be needed. We actually think that the response and recovery to the crises impacts QoL, not the other way round. However, we acknowledge that a reversed causality might also be possible. We hypothesize that the economic crisis in Portugal and Spain might explain some of our results, but we have no data on the “how”. An answer to this question would have to come from longitudinal studies. For example, in the case of Switzerland, it was observed that the global recession of the early 2000s had a profound effect on the decline in the quality of life of vulnerable groups (Simona-Mousa & Ravazzini, 2019). Moreover, our study is not based on stochastic or predictive data and heteroscedasticity models have not been used which could show the economic impact due to QoL. We will take your proposal in consideration for future papers.

Reference:

- Simona-Moussa, J., & Ravazzini, L. (2019). From One Recession to Another: Longitudinal Impacts on the Quality of Life of Vulnerable Groups. Social Indicators Research, 142(3), 1129-1152.  https://doi.org/10.1007/s11205-018-1957-5

We have discussed these issues adding the following text to the 3rd paragraph of the Discussion:

“A comparative European study by country evaluated the changes in QoL between 2007 and 2012, and revealed the negative impact of the economic crisis on the standard of living, trust in institutions and other aspects of health conditions, although it did not find statistically significant changes in relation to the global QoL indicator, highlighting that “the most marked changes have been identified in the single groups of QoL rather than on the overall index QoL: this is because the opposite direction of some changes have tended to cancel out when we consider the overall average effect” [33].”

Line 255 parentheses are used rather than brackets for the reference.

Answer: This reference has been corrected:

“Depression is also a longitudinal predictor of QoL [38].”

Authors may want to further discuss potential reasons why they did not find a significant effect of socio-economic status considering the impact of the economic crisis in Portugal and especially in Spain. The author’s point on psychosocial and health variables relative importance is well taken although it stands to reason health and psychosocial variables may have also been impacted by the economic crisis and therefore may not have had as strong of a controlling effect on the impact of financial security. This may be worth exploring further.

Answer: We actually use wealth as a proxy for socioeconomic status; this was now corrected throughout the manuscript. The reason for not finding an effect as statistically significant may be diverse, including lack of statistical power for this effect, the use of other variables that explain a higher proportion of variance, or that a variable could be acting as a mediators instead of a direct effect. In our study, wealth included value of the main residence, which was less affected by the economic crisis than other indicators such as income. We agree with the reviewer’s explanation, and this was added to the discussion:

“Our study did not find a direct effect of wealth on QoL change, even though Portugal and Spain were influenced by the economic crisis during the study period. The economic crisis directly affected wealth, as well indirectly health and psychosocial variables.”

Country and social participation interaction in Spain, what are the potential implications of this on policy specific to Spain?

Answer: The following sentence was added:

“This study showed an interaction between QoL, participation and country, as evidenced by the association of participation with increased QoL only in Spain. Policies aimed at increasing the QoL of older adults in Spain should promote participation activities and enable places where these can occur.“

Overall, authors wrote a thorough and thoughtful manuscript on an interesting and relevant data analysis project. Authors carried out thorough and responsible data analyses and wrote a clear and concise manuscript.

Answer: We thank the reviewer for this comment.

Reviewer 3 Report

The paper  focuses on the analyze of the determinant of quality of life of older people in three European countries.

I find this topic very interesting, useful and up to date in many contexts, e.g.: social, demographic, economic, administrative,

The applied research methods are adequate to the problem. 

The article contains almost the appropriate structure. The article has been correctly divided into relevant sections, and their content coincides with their titles. However, there is no section devoted to the literature analysis and basically there is no "conclusions" section. It should be supplemented.

Footnotes and bibliography are correctly formulated.

The language of the article is mature, correct, adequate.

However, some parts of the article have to be strengthened:

  • add section devoted to literature review;
  • in my opinion, the number of references is not sufficient. Authors should supplement the article with more references. When the Authors supplement the article with part devoted to literature review, the number of references should also increase;
  • the issue of healthy aging was very briefly discussed. It has not been discussed what factors contribute to healthy aging and what it is. Giving a few definitions did not cover the topic;
  • I believe that the justification for the selection of countries where the study was conducted needs to be deepened. These are not the only countries where there is a problem of depopulation and aging of society. An example is Poland. It is understandable to show the example of Spain and Portugal together. However, Sweden has a different specificity in a broader context: population density, topography, climate, economy, etc. The sample selection should be better justified;
  • Another, not fully clarified, issue is the selection of the years from which the data from the survey was selected;
  • there is also a lack of justification why these and not other indicators were selected for the research. Why this particularly factors was taken into concideration? I miss a deeper explanation of the selection of factors for analysis.
  • apart from the abstract, the purpose of the article/research was not specified
  • no hypothesis was given;
  • Two sentences that make up the entirety of the ending are certainly not enough. Basically we can say that the ending part is missing...

Author Response

The paper  focuses on the analyze of the determinant of quality of life of older people in three European countries. I find this topic very interesting, useful and up to date in many contexts, e.g.: social, demographic, economic, administrative. The applied research methods are adequate to the problem. 

The article contains almost the appropriate structure. The article has been correctly divided into relevant sections, and their content coincides with their titles. However, there is no section devoted to the literature analysis and basically there is no "conclusions" section. It should be supplemented.

Answer: A comprehensive literature review is out of the scope of this paper, at least in the sense other larger papers do. The second introduction paragraph revises important studies directly related to our study goal, as now stated. In addition, the literature has been revised and pertinent references have been added where appropriate. The Conclusions section, which has been lengthened, is presented at the end of the discussion:

“In conclusion, determinants of QoL changes are multidimensional, thus calling for a multidisciplinary and multilevel approach. When comparing the QoL level of Sweden, Portugal and Spain, we observe a North-South gradient. Also, the observed change of QoL level in the study period was different by country and is probably related to a differential response to the economic crisis in the previous period. Our study showed that better physical, emotional and functional health in old age is related to an increase in QoL in Portugal, Spain, and Sweden, while a higher social participation is related to an increase in QoL only in Spain. Particularly in Spain, interventions that offer social participation activities and enable places for it, might be of great benefit for older adults’ QoL. This study suggests the importance of taking into account the influence that health and social participation have according to the context of each country when developing public health policies to promote QoL among European older people.”

Footnotes and bibliography are correctly formulated. The language of the article is mature, correct, adequate.

However, some parts of the article have to be strengthened:

  • add section devoted to literature review;

Answer: As explained above, a comprehensive literature review is out of the scope of this paper. The second introduction paragraph revises important studies directly related to our study goal, a now stated. In addition, the literature has been revised and pertinent references have been added where appropriate.

  • in my opinion, the number of references is not sufficient. Authors should supplement the article with more references. When the Authors supplement the article with part devoted to literature review, the number of references should also increase;

Answer: We have added more references to the text, it has now 54. Most medical journals have a limit of 30-40 references for empirical papers. 

  • the issue of healthy aging was very briefly discussed. It has not been discussed what factors contribute to healthy aging and what it is. Giving a few definitions did not cover the topic;

Answer: Two frameworks are used in this work: active and healthy ageing. Both are briefly presented and referenced to the complete texts, where its contributing factors are fully discussed. However, our study does not describe the determinants of healthy ageing. Instead, it focuses on healthy and active ageing dimensions as determinants of QoL.

  • I believe that the justification for the selection of countries where the study was conducted needs to be deepened. These are not the only countries where there is a problem of depopulation and aging of society. An example is Poland. It is understandable to show the example of Spain and Portugal together. However, Sweden has a different specificity in a broader context: population density, topography, climate, economy, etc. The sample selection should be better justified;

Answer: As explained to Reviewer 2, there was an interest in comparing Northern and Southern countries such as Sweden, Portugal and Spain according to the evidence of a gradient found in previous studies using SHARE data. Moreover, the research team is composed by researchers from the three countries that could provide insight and draw conclusions based on the socio-economic and cultural contexts of the participating countries. We agree with the reviewer that further studies are called upon, including Eastern-Europe countries such as Poland. We have added this to the study limitations:

“It would be very interesting to widen the geographical scope of the study, especially to include Eastern-Europe countries.”

  • Another, not fully clarified, issue is the selection of the years from which the data from the survey was selected;

Answer: As explained in the methods section, these two waves were chosen since they provided data about the measures of interest for the three countries. We selected the most recent waves available by the time we did the analysis, that had data for the three participating countries. For example, Portugal has not data in wave 5, so we could not include it.

  • there is also a lack of justification why these and not other indicators were selected for the research. Why this particularly factors was taken into concideration? I miss a deeper explanation of the selection of factors for analysis.

Answer:  The selection of the main components was guided by the active and healthy ageing frameworks. This is explained in the methods sections:

“Five experts in the field of ageing research reviewed the SHARE databases to identify indicators of health, social participation and lifelong learning, the main predictors related to active and healthy ageing examined in the present study. Along with security, these dimensions correspond to the pillars of the active ageing framework [1]. In addition, health and participation are the core of the healthy ageing model [2].”

  • apart from the abstract, the purpose of the article/research was not specified

Answer: The purpose of the study is in the before-last paragraph of the Introduction:

“The main objective of this article was to analyze how health, social participation and lifelong learning indicators predict changes in QoL in older people, and to ascertain if such predictors differ from one country to another”.

However, according to the reviewers’ comments and for clarity purposes, we have added the following text:

“Using data from the Survey of Health, Ageing and Retirement in Europe (SHARE), this study hypothesizes that there will be significant differences in QoL between two waves of the survey and in the three studied countries (Portugal, Spain and Sweden). In addition, we will explore the determinants of change in QoL in the three countries.”

  • no hypothesis was given;

Answer: As presented above, hypotheses are now added:

“Using data from the Survey of Health, Ageing and Retirement in Europe (SHARE), this study hypothesizes that there will be significant differences in QoL between two waves of the survey and in the three studied countries (Portugal, Spain and Sweden). In addition, we will explore the determinants of change in QoL in the three countries.”

  • Two sentences that make up the entirety of the ending are certainly not enough. Basically we can say that the ending part is missing...

Answer: The conclusions have been expanded and now read as follows:

“In conclusion, determinants of QoL changes are multidimensional, thus calling for a multidisciplinary and multilevel approach. When comparing the QoL level of Sweden, Portugal and Spain, we observe a North-South gradient. Also, the observed change of QoL level in the study period was different by country and is probably related to a differential response to the economic crisis in the previous period. Our study showed that better physical, emotional and functional health in old age is related to an increase in QoL in Portugal, Spain, and Sweden, while a higher social participation is related to an increase in QoL only in Spain. Particularly in Spain, interventions that offer social participation activities and enable places for it, might be of great benefit for older adults’ QoL. This study suggests the importance of taking into account the influence that health and social participation have according to the context of each country when developing public health policies to promote QoL among European older people.”

Round 2

Reviewer 3 Report

Thank you very much to the authors for supplementing the article as a result of the reviews presented.Leaving aside the positive issues that have already been discussed in the previous review, I would like to focus on a few remarks for improvement:

·         The literature part has not been added to the paper, but only the introduction has been extended. It doesn't seem right. The introduction itself should be expanded, and a literature review section should be added additionally

·         The number of publications has been increased, but only by 6 items. This is not a significant qualitative change. In my opinion, these parts of the article have not been corrected

·         It can be concluded that the issues of healthy aging have been supplemented

·         The article was supplemented with explanations why three countries were selected for the research sample. This is not an in-depth explanation, but it gives some picture of the situation and justifies the sample selection. Similarly, the authors approached the explanation of the years, from which the data from the survey was selected

  • still seems that there is also a lack of justification why these and not other indicators were selected for the research. Why this particularly factors was taken into concideration? I miss a deeper explanation of the selection of factors for analysis.
  • The article was supplemented with the hypothesis formulated in the introduction. Please, let the authors refer to it at the end of the article
  • apart from the abstract, the purpose of the article/research was still not specified
  • The discussion part has been significantly updated

·         However, Conclusions part still needs to be supplemented

Author Response

Thank you very much to the authors for supplementing the article as a result of the reviews presented. Leaving aside the positive issues that have already been discussed in the previous review, I would like to focus on a few remarks for improvement:

·         The literature part has not been added to the paper, but only the introduction has been extended. It doesn't seem right. The introduction itself should be expanded, and a literature review section should be added additionally

·         The number of publications has been increased, but only by 6 items. This is not a significant qualitative change. In my opinion, these parts of the article have not been corrected

Answer: As explained in the first revision, a comprehensive literature review is out of the scope of this paper. The second introduction paragraph revises important studies directly related to our study goal, as now stated. In addition, the literature has been revised and pertinent references have been added where appropriate. We have added more references to the text, it has now 54, reflecting an increase by one eight (12.5%). Most medical journals have a limit of 30-40 references for empirical papers.  In 6 papers co-authored by our team and published in this same journal, the mean reference number was 42 (range: 32-48). In addition, the Introduction has increased by almost 300 words, following the previous reviewers’ suggestions, and it cites 18 references. The literature review is included in the Introduction, and further literature is reviewed in the Discussion. According to the journal’s Instruction for authors, the following are the required sections: “Author Information, Abstract, Keywords, Introduction, Materials & Methods, Results, Conclusions, Figures and Tables with Captions, Funding Information, Author Contributions, Conflict of Interest and other Ethics Statements.” It does not include a “literature review” section.

  • It can be concluded that the issues of healthy aging have been supplemented

    ·         The article was supplemented with explanations why three countries were selected for the research sample. This is not an in-depth explanation, but it gives some picture of the situation and justifies the sample selection. Similarly, the authors approached the explanation of the years, from which the data from the survey was selected

Answer: As explained in the first revision, there was an interest in comparing Northern and Southern countries such as Sweden, Portugal and Spain according to the evidence of a gradient found in previous studies using SHARE data. Moreover, the research team is composed by researchers from the three countries that could provide insight and draw conclusions based on the socio-economic and cultural contexts of the participating countries. The two waves were chosen since they provided data about the measures of interest for the three countries. We selected the most recent waves available by the time we did the analysis, that had data for the three participating countries. For example, Portugal has not data in wave 5, so we could not include it.

It is a subjective opinion to state that the explanation provided for selecting the three countries “is not an in depth-explanation”, which we disagree. In the third Introduction paragraph, we offer theoretical arguments to choose these countries. The pragmatic explanation, stated above, is also explained to the reviewer.

still seems that there is also a lack of justification why these and not other indicators were selected for the research. Why this particularly factors was taken into concideration? I miss a deeper explanation of the selection of factors for analysis.

Answer: The selection of the main components was guided by the active and healthy ageing frameworks. This is explained in the methods sections:

“Five experts in the field of ageing research reviewed the SHARE databases to identify indicators of health, social participation and lifelong learning, the main predictors related to active and healthy ageing examined in the present study. Along with security, these dimensions correspond to the pillars of the active ageing framework [1]. In addition, health and participation are the core of the healthy ageing model [2].”

This reviewer’s comment is copied from the previous review, despite our answer. A “deeper explanation” is a personal, unfounded opinion. The explanation we offer is actually theoretical based in two WHO models (active and healthy ageing frameworks). In addition, the choice of indicators was limited to the ones available in the SHARE database. All this is stated in the second paragraph of section 2.2. Measurements.

The article was supplemented with the hypothesis formulated in the introduction. Please, let the authors refer to it at the end of the article

Answer: We have added a comment on the hypotheses at the beginning of the Discussion.

apart from the abstract, the purpose of the article/research was still not specified

Answer: As explained in the first revision, the purpose of the study is in the before-last paragraph of the Introduction:

“The purpose of this article was to analyze how health, social participation and lifelong learning indicators predict changes in QoL in older people, and to ascertain if such predictors differ from one country to another. Using data from the Survey of Health, Ageing and Retirement in Europe (SHARE), this study hypothesizes that there will be significant differences in QoL between two waves of the survey and in the three studied countries (Portugal, Spain and Sweden). In addition, we will explore the determinants of change in QoL in the three countries.”

The study purpose or main question is also presented in the first line of the abstract and statistical analysis section (“to analyze the variables associated with the change in QoL”) and it is answered in the first paragraph of the discussion and conclusions.

The discussion part has been significantly updated
·         However, Conclusions part still needs to be supplemented

Answer: Following the first review, the conclusions were expanded and its word count is now close to double:

“In conclusion, determinants of QoL changes are multidimensional, thus calling for a multidisciplinary and multilevel approach. When comparing the QoL level of Sweden, Portugal and Spain, we observe a North-South gradient. Also, the observed change of QoL level in the study period was different by country and is probably related to a differential response to the economic crisis in the previous period. Our study showed that better physical, emotional and functional health in old age is related to an increase in QoL in Portugal, Spain, and Sweden, while a higher social participation is related to an increase in QoL only in Spain. Particularly in Spain, interventions that offer social participation activities and enable places for it, might be of great benefit for older adults’ QoL. This study suggests the importance of taking into account the influence that health and social participation have according to the context of each country when developing public health policies to promote QoL among European older people.”

We disagree with the one-sided view that the conclusions need to be further expanded. We have summarized the main findings and recommendations in a brief, concrete way. Moreover, we have answered the main purpose and hypotheses of the study. To add more words without a specific purpose would distract the reader from the main message we intent to convey.
